# Elevated Temperature Tensile Creep Behavior of Aluminum Borate Whisker-Reinforced Aluminum Alloy Composites (ABOw/Al–12Si)

**DOI:** 10.3390/ma14051217

**Published:** 2021-03-04

**Authors:** Yameng Ji, Yanpeng Yuan, Weizheng Zhang, Yunqing Xu, Yuwei Liu

**Affiliations:** 1School of Mechanical Engineering, Beijing Institute of Technology, Beijing 100081, China; jiyameng08@163.com (Y.J.); zhangwz@bit.edu.cn (W.Z.); 2Shandong Binzhou Bohai Piston Co., Ltd., Binzhou 256602, China; xu.yunqing@163.com; 3School of Mechanical Electronic and Information Engineering, China University of Mining and Technology, Beijing 100083, China; yuweiliu1@126.com

**Keywords:** Al_18_B_4_O_33_ whisker-reinforced Al–12Si composite, creep behavior, stress exponent, load transfer, creep fracture

## Abstract

In order to evaluate the elevated temperature creep performance of the ABOw/Al–12Si composite as a prospective piston crown material, the tensile creep behaviors and creep fracture mechanisms have been investigated in the temperatures range from 250 to 400 °C and the stress range from 50 to 230 MPa using a uniaxial tensile creep test. The creep experimental data can be explained by the creep constitutive equation with stress exponents of 4.03–6.02 and an apparent activation energy of 148.75 kJ/mol. The creep resistance of the ABOw/Al–12Si composite is immensely improved by three orders of magnitude, compared with the unreinforced alloy. The analysis of the ABOw/Al–12Si composite creep data revealed that dislocation climb is the main creep deformation mechanism. The values of the threshold stresses are 37.41, 25.85, and 17.36 at elevated temperatures of 300, 350 and 400 °C, respectively. A load transfer model was introduced to interpret the effect of whiskers on the creep rate of this composite. The creep test data are very close to the predicted values of the model. Finally, the fractographs of the specimens were analyzed by Scanning Electron Microscope (SEM), the fracture mechanisms of the composites at different temperatures were investigated. The results showed that the fracture characteristic of the ABOw/Al–12Si composite exhibited a macroscale brittle feature range from 300 to 400 °C, but a microscopically ductile fracture was observed at 400 °C. Additionally, at a low tensile creep temperature (300 °C), the plastic flow capacity of the matrix was poor, and the whisker was easy to crack and fracture. However, during tensile creep at a higher temperature (400 °C), the matrix was so softened that the whiskers were easily pulled out and interfacial debonding appeared.

## 1. Introduction

With the continuous improvement of diesel engine-strengthening level, high speed and high-power density have become the main trend of diesel engine development in recent years. The strength and reliability of diesel engine combustion chamber parts are required to be higher [1]. As the key part of the transmitting force and torque, piston is in direct contact with high-temperature gas. The strength of the piston material is greatly reduced due to excessive temperature, and its reliability becomes an important factor in restricting the safe operation of the whole machine [2]. The local strengthening technology is to use a reinforced aluminum matrix composite with high strength in the bowl edge of the piston crown. The application of composite materials increases the strength of the piston at elevated temperatures and then improves greatly the creep resistance of the piston crown [3].

Aluminum matrix composites reinforced by aluminum borate whiskers (Al_18_B_4_O_33_ denoted by ABOw) are attractive for a wide application in automotive engine piston manufacture because of their good wear resistance, high specific strength, and excellent elevated temperature stability [4,5,6,7]. As we know, the maximum temperature of the piston is located on the combustion chamber edge of the piston crown, and the temperature of the dangerous local point after strengthening can reach more than 400 °C [8,9]. Therefore, the high temperature performance of the material will drop greatly.

The elevated-temperature creep property of materials is a significant index to evaluate the high-temperature performance of materials. On the one hand, the high-temperature creep data of materials can serve the practical engineering application of materials; on the other hand, they provide certain supporting theories for the high-temperature creep research of materials. To determine the high-temperature properties of aluminum matrix composites and expand their application range, many scholars have studied the elevated temperature creep of aluminum matrix composites in recent years, which shows that the composites have excellent high-temperature creep resistance. Peng et al. [10] studied the high-temperature strengthening mechanism of discontinuous metal matrix composites and observed that the presence of whiskers decreases the creep rate by about two orders of magnitude. They suggested that the dominant deformation mechanism in the discontinuous metal matrix composites was climb-controlled dislocation creep at elevated temperatures. Ma et al. [11] reported higher creep apparent stress exponents and apparent activation energies in discontinuous reinforced aluminum matrix composites. It was shown that the stress exponents could be rationalized to about five by incorporating a threshold stress into the analysis. The threshold stress was introduced to explicate the creep behavior of these aluminum matrix composites [12]. Furthermore, Ji et al. [13] justified the need for a threshold stress, in the light of the much higher stress exponents and activation energies obtained, compared to those of pure aluminum. The addition of whiskers or particulates could markedly increase the creep resistance of aluminum alloys [14].

Nevertheless, so far, only limited information is available about high-temperature properties (especially high-temperature creep behavior) of ABOw-reinforced Al matrix composites and the temperature influence on fracture mechanisms in the ABOw/Al–12Si composite system. The aim of the current study is to investigate the creep deformation properties of AlBOw-reinforced Al matrix composites and to compare them with the creep deformation properties of the unreinforced Al–12Si alloy. The microstructure evolution fracture mechanism during creep is discussed in order to give a thorough understanding of the operating creep mechanism.

## 2. Material and Experimental Procedures

The materials used in this study are the Al–12Si alloy (as shown in Table 1) and the 23 vol.% ABO whiskers-reinforced Al–12Si composite, manufactured by the squeeze-casting technique. The ABOw with a diameter of 0.5–1.5 µm and a length of 10–30 µm and the Al–12Si aluminum alloy were used as a reinforcement and matrix. Figure 1 is the SEM (JEOL, Tokyo, Japan) fractograph of the ABOw/Al–12Si composite. It can be seen that the distribution of aluminum borate whiskers was homogeneous and random in the composite and without matrix cast defect. For comparison, the unreinforced Al–12Si piston alloy was also prepared by the squeeze-casting technique. 

As shown in Figure 2a,b, obvious bulk primary silicon, aluminum, and intermetallic compounds can be seen in the aluminum base alloy on the left side, most of which were hexagonal, petal, and dendritic in shape, respectively. The grain size of the composite was smaller than that of the Al–12Si alloy, which indicated that the addition of whiskers made α-Al and silicon particles further refined. The above materials were obtained from BoHai Piston Corporation, Binzhou, China.

For the metal matrix composites, the creep time was longer than that of the matrix alloy. In order to obtain a stable second creep stage in a reasonable time, relatively higher stresses should be applied at each temperature during the creep test. The rapid creep test method is usually used to analyze the creep behavior of composites. Therefore, it is necessary to accurately measure the tensile strength of the material. When the stress acting on the material is close to 0.8 times of the strength of tensile, the material will produce rapid creep under high temperature. Polished specimens were tested on a servo-hydraulic machine (SINOTEST, Changchun, China) at a constant strain rate of 1 × 10^−4^ s^−1^ in a temperature range from 200 to 350 °C. Creep tests were performed on a RDJ30 creep testing machine (SINOTEST, Changchun, China) at Sinotest Equipment Co., Ltd. in Changchun City, Jilin Province, China. The dimension of the tensile specimen is shown in Figure 3, and the size of the tensile creep specimen was 25 mm in length and 5 mm in diameter. According to the test results of tensile strength, the creep temperatures were 250, 300, 350 and 400 °C, respectively, and the stress ranged from 70 to 230 MPa. The tensile fractographs of composites were examined by SEM (JEOL, Tokyo, Japan) for gaining an understanding of the failure mechanisms. The selected samples near the creep fracture were examined along the longitudinal direction with Transmission Electron Microscope (TEM) (JEOL, Tokyo, Japan). The specimens for TEM (JEOL, Tokyo, Japan) were mechanically ground to uniform slices of about 200 µm in thickness and further thinned to 100 nm by twin-jet electro polishing. The TEM (JEOL, Tokyo, Japan) examinations were conducted on JEM-2010 operating at 80–200 kV.

## 3. Results and Discussion

### 3.1. Mechanical Properties

The tensile properties of the Al–12Si alloy and the ABOw/Al–12Si composite at different temperatures are listed in Table 2. From Table 2, the results showed the Young’s modulus and ultimate tensile strength (UTS) of the ABOw/Al–12Si composite were significantly higher than those of Al–12Si alloy, which indicated the effect of whisker was greatly advanced. From room temperature to 350 °C, the UTSs of the Al–12Si alloy and the ABOw/Al–12Si composite decreased by 69% and 44%, respectively, which indicated that the ABOw/Al–12Si composite had excellent high-temperature stability [15]. Moreover, the Young’s modulus indicates the rigidity of the material. The greater the Young’s modulus is, the less likely it is to deform. In addition, it is largely decided by the load transfer capacity of the material, indicating that the addition of whiskers has excellent load transfer capacity for the Al–12Si alloy. Furthermore, the elongation of the Al–12Si alloy was far away greater than that of the ABOw/Al–12Si composite from 25 to 350 °C, which revealed that the plastic properties of composites were poor compared with those of the aluminum alloy. The elongations of the ABOw/Al–12Si composite specimens were only from 0.61% to 1.25% in the 25–350 °C temperature range. It is well known that reinforced whiskers, as barriers to hinder the dislocation movement, limit the deformation of a matrix under applied loads, resulting in the decrease of plasticity. Fei [16] did researches on the high-temperature creep deformation of the Al_18_B_4_O_33_ whisker-reinforced 8009 Al composite. They presented that the addition of the whiskers does not evidently affect the size and the distribution of the dispersoids but there are more dislocations around the whiskers, indicating the strengthening effects of the whiskers.

### 3.2. Tensile Creep Properties

Creep tests were performed at a temperature within the range of 50–400 °C under different applied stresses. The creep tests of most specimens were terminated until the specimens fractured, and few specimens were stopped in the defined steady state creep region. Figure 4 shows creep tensile curves expressed as strain against the dwell time of the ABOw/Al–12Si composites tested at 250, 300, 350 and 400 °C, respectively. According to these strain–time curves, the creep curve has an obvious initial creep and a stabilized creep stage from the selected temperature and stress range. It can be seen that the creep rate decreased drastically in the deceleration creep stage, followed by a stationary creep stage in which the strain increased linearly with time. However, a great number of creep experiments [17,18] on the unreinforced aluminum matrixes revealed that their strain–time curves show three typical creep stages, named initial creep stage, steady state creep stage, and accelerated creep stage. Apparently, the ABOw/Al–12Si composites had a long secondary creep stage.

For the whisker-reinforced composites, dislocations accumulated at the interface at the early stage of creep, and the dislocation density increased gradually, forming a work hardening area around the whisker. When the dislocation line passed through the whisker, the dislocation ring formed. When the dislocation ring moved along the whisker to the end, the dislocation ring shrank and annihilated. As described in [19,20], the existence of the whisker greatly restricts the dislocation movement, hinders the deformation of the base alloy and then strain hardening occurs in the material, resulting in the strain change in the initial creep stage. Due to the strain mismatch between the whisker and the matrix, the stress concentration area was formed in the matrix alloy around the whisker. When the strain of whisker achieved the fracture strain critical value in the composite, the whisker broke, which made the dislocation slip and climb to the end of the fiber quickly and shrink. Then, the stress on the whisker can be relaxed rapidly. The creep of the composite reached a stable stage, when the relaxation and the loading process reached the balance, resulting in a constant creep rate (Figure 5).

However, for the composite materials, there was no obvious accelerated creep stage in the creep fracture samples, which indicated that the creep fracture was sudden and the composite materials had obvious brittle fracture characteristics. Moreover, there was a significant transition behavior between the primary creep stage and the stationary creep stage, which is consistent with the description in [21]. Furthermore, it is evident that the stationary creep stage arose at a shorter period and the creep resistance decreased gradually, with the increase of creep stress under a constant temperature. The steady-state creep rate could be obtained through fitting the steady-state creep stage curve with the least square method, and the experiment results are listed in Table 3. The stationary creep strain rates were calculated at different creep temperatures, and creep stresses are listed. From Table 3, it can be seen that the stationary creep rate generally increased with increasing temperature at any given temperature. 

In the steady-state creep stage, in order to facilitate the comparison, the characteristic creep strain curves of the unreinforced aluminum alloy and the ABOw/Al–12Si composite at 350 °C and 70 MPa are given in Figure 6. The comparison showed that the aluminum material had a large amount of creep deformation, which showed a short period of the primary creep stage at the beginning of the experiment followed by the accelerated creep stage. However, the creep curve of the composite was still at the slow stationary creep stage. Moreover, the creep deformation of the composite was far below than that of the Al–12Si alloy, which showed that the composite had a strong creep resistance compared with the aluminum matrix under the same load and same temperature conditions. The creep rate of the aluminum alloy material was calculated to be 288.71 × 10^−8^ s^−1^. It can be seen from Table 1 that the minimum creep rate of the aluminum borate whisker composite was three orders of magnitude higher than that of the aluminum alloy under the same testing condition. For instance, the aluminum borate whisker-reinforced composite exhibited a stationary creep period for 38 h at 350 °C and 70 MPa, whereas the unreinforced aluminum alloy had already undergone creep fracture after 1.5 h. These results indicated that with the creep resistance of the composite has been greatly improved by adding whiskers. Whiskers affected the growth and refinement of the aluminum alloy grains as seen in Figure 1 and Figure 2a. The reason is that the addition of ABO whiskers increased the number of heterogeneous nucleation in the solidification process of the Al–12Si alloy and hindered the grain growth of the alloy, thus refining the structure of the alloy. The generation and disappearance of dislocations in the fine grains mostly occurred at the interface between the whiskers and the matrix, so the creep resistance of the composite has been improved. This phenomenon is the same as the comparison of the creep rates of the SiC/8009 composite with those of the 8009 alloy [14]. 

### 3.3. Creep Activation Energy and Stress Exponent

The creep behavior of a material is mainly decided by the characteristics of the material itself and external conditions. Moreover, the steady-state rate is a significant parameter to measure the creep performance of the alloy, which depends on both the creep temperature and the applied load. According to the present study, the activation energy *Q* and the creep stress exponent *n* are significant parameters to understand the creep mechanism in the creep constitution equation [22]. The equation shown as Equation (1) reflects the relationship between these two parameters and the creep rate [23,24]:(1)$$\dot{\varepsilon }=\frac{AGbD_{o}}{kT}{\left(\frac{\sigma }{G}\right)}^{n}\exp(-\frac{Q}{RT})$$
where *G* is the shear modulus, σ is the applied stress, *b* is the Burgers vector, *T* is the absolute temperature, $\dot{\varepsilon }$ is the minimum creep rate, $D_{o}$ is the diffusion coefficient, *k* is the Boltzmann constant, *Q* is the activation energy for creep, *n* is the stress exponent, *R* is the universal gas constant and *A* is a dimensionless constant.

According to Equation (1), the dependence of the stress on the creep strain rate can be obtained from the stress exponent *n*, defined as:(2)$$\ln\dot{\varepsilon }=\ln A_{1}+n\ln\sigma$$
where $A_{1}$ is the material characteristics parameter. 

The value of stress exponent n originates from a straight-line slope, which was calculated by fitted data from ln$\dot{\varepsilon }$ and lnσ in Figure 7. The values of the stress exponent at different temperatures for the ABOw/Al–12Si composite and the Al–12Si alloy are summarized in Table 4.

Stress exponent values are in the range 4.03–6.02 for the ABOw/Al–12Si composite, depending on the test temperature. Similar *n* values have been reported in [25]. For *n* = 4–6, the creep was controlled by dislocation climb [26]. It was shown that the dislocation movement in the creep process should conform to the dislocation climbing mechanism. In addition, the stress exponents decreased significantly with decreasing of the creep temperature. This is because the higher the creep temperature was, the more sensitive the creep rate was to the creep stress. 

The creep activation energy *Q* can be given as the following:(3)$$Q=R\frac{\ln(\varepsilon_{1}^{’}/\varepsilon_{2}^{’})}{\frac{1}{T_{2}}/\frac{1}{T_{1}}}$$

The apparent creep activation energy for the creep of the aluminum borate whisker-reinforced composite was calculated to be 148.75 kJ/mol under the stress condition of 70 MPa at temperatures of 350 and 400 °C, which approached the activation energy for lattice self-diffusion in aluminum. Dislocation creep or power law creep mainly occurred by the movement of dislocations, which is explained by the dislocation climb mechanism [25]. The climb of dislocations became the controlling mechanism at high temperatures with n values in the range 4–6 and the activation energy that is equal to the activation energy for lattice self-diffusion [27]. In our present research, a power-law relationship with n values in the range of 4.03–6.02 was valid in the temperature range of 300–400 °C. Furthermore, the addition of AlBO whiskers could greatly improve the creep resistance of the composites but not evidently affected the stress exponent and the creep activation energy compared with the unreinforced aluminum alloy. This phenomenon is the same as the creep behavior in SiC whiskers [14].

To reveal the operative deformation mechanism, the dislocation substructure developed in the specimens after the creep experiment was examined by TEM. The reason for creep is that dislocations can overcome the obstacles such as dislocation stacking and move and vacancies can migrate in a directional way, with the help of the thermal activation energy provided by the external load and temperature in the high temperature environment.

Figure 8 is the subgrain caused by dislocation under the applied stress 70 MPa at 350 °C in the sample. Figure 8 is the structural morphology of dislocations near the whisker, which shows that dislocations were accumulated near the whisker. The dislocations were tangled around the precipitates, and it was difficult to rearrange into dislocation walls by bypassing one obstacle after another. 

From the above observations, under the action of high-temperature tensile creep, a large number of dislocations were generated and hindered by the whiskers, intertwining and pinning each other, and the internal hardening of the material was strengthened, so that the creep strain of the composite was reduced. The main reason is that the whiskers replaced the metal compounds, which were easy to soften at high temperature at the grain boundary and the pinning effect of reinforcing phase prevented the occurrence of grain boundary cross slip and dislocation climb at high temperature.

### 3.4. Threshold Stress Analysis

It is well documented that the creep resistance of the ABOw/Al–12Si composite was higher than that of the Al–12Si alloy in this study. The creep behavior of discontinuous aluminum and aluminum alloy matrix composites was recently interpreted by the threshold stress approach [13]. The creep deformation is not driven by the applied stress, but the effective stress, $\sigma -\sigma_{0}$. Therefore, Equation (1) can be rewritten as:(4)$$\dot{\varepsilon }=\frac{AGbD}{kT}{\left(\frac{\sigma -\sigma_{o}}{G}\right)}^{n}$$
where $\sigma_{o}$ is the threshold stress and *n* is the effective stress exponent.

The present experimental data of the ABOw/Al–12Si composite were examined by using an approach previously reported to obtain the value of the threshold stress [28]. Through the present experimental data, the threshold stress can be calculated accurately using Equation (4). The creep data of ${\dot{\varepsilon }}^{1/n}$ and σ were plotted on double linear coordinates at each temperature. By extrapolating the data linearly to the zero strain rate, the value of threshold stress for static creep can be obtained. For n = 4, a linear relationship for the present composite was obtained at three temperatures in Figure 9. Therefore, the threshold stress values were 37.41, 25.85 and 17.36 MPa at 300, 350 and 400 °C from this plot, respectively. It can be seen that the value of threshold stress decreased with the increase of temperature. Table 5 lists all data of the creep threshold stresses in the ABOw/Al–12Si composite.

The creep threshold stress mainly originates from the interaction between movable dislocations and precipitates in a matrix [15]. In recent years, there have been many studies on the threshold stress behavior in discontinuous particle-reinforced composites. Gonzalez-Doncel and Sherby [29] also reported that the origin of the threshold stress was linked to SiC particles or SiC whiskers in discontinuous SiC/Al composites. The influence of aluminum borate whiskers in aluminum alloys on creep behavior are not the same as that of particle-reinforced aluminum alloys.

Ji et al. [13] suggested the resistance of a particle-reinforced aluminum alloy was originated from particles to the deformation of a matrix and the pinning of precipitated phases then to dislocation. However, in our present research, under the action of high-temperature tensile creep, a large number of dislocations were generated and hindered by whiskers, which can intertwine and pin each other. The internal hardening of the material was strengthened, so that the creep strain of the reinforced composite was reduced.

Thus, a great number of aluminum borate whiskers blocked the motion of dislocation. While the external stress exceeded to the threshold stress, the creep occurred. Due to the strain mismatch between the whisker and the matrix, the stress concentration zone was formed in the matrix alloy around the whisker under the action of an external load. When the strain of the whisker in the composite reached the fracture strain value, the whisker fractured. The dislocation could quickly slip and climb to the end of the fiber, then shrank and disappeared, which made the dislocation movement continue. 

### 3.5. Contribution of Whiskers by Load Transfer

The high creep resistance may be attributed to the following: (a) the role of the whiskers in slowing the motion of dislocations via viscous glide and climb [30] and (b) the load transfer between the matrix and large incoherent particles [31]. According to the above results in this paper, the addition of the whiskers can greatly improve the creep resistance of the composite but do not fully explain the abnormally low creep rate of the composite compared to the matrix (Figure 2). The existence of rigid reinforcement undertook most of the applied load, which made the creep resistance of the composite increase [32,33]. The load-transfer mechanism can now be examined for this phenomenon. A continuum mechanics model of Kelly [34] predicts the creep property of discontinuous fiber-reinforced composites. For this composite, Equation (5) can be used to predict the creep rate:(5)$${\dot{\varepsilon }}_{c}={\dot{\varepsilon }}_{m}{\left[(1-V_{f})+\lambda_{1}\lambda_{2}{\left(\frac{L}{d}\right)}^{\frac{n+1}{n}}V_{f}\right]}^{-n}$$
where ${\dot{\varepsilon }}_{c}$ is the composite creep rate, L/d is the aspect ratio of the reinforcement, ${\dot{\varepsilon }}_{m}$ is the matrix creep rate at the same temperature and stress, n is the matrix stress exponent, $V_{f}$ is the volume fraction of the reinforcement, $\lambda_{1}$ and $\lambda_{2}$ are the whisker orientation and whisker length factors, respectively, and β is the load transfer coefficient decided by:(6)$$\beta ={\left(\frac{2}{3}\right)}^{\frac{1}{n}}(\frac{n}{2n+1}){\left[{\left(\frac{2\sqrt{3}}{\pi }V_{f}\right)}^{-\frac{1}{2}}-1\right]}^{-\frac{1}{n}}$$

For the present ABOw/Al–12Si composite, using *n* = 4.18, $V_{f}$ = 0.24, and L/d = 25, Equation (6) gives the value of β as 0.415. During the extrusion process, progressive and continuous changes in whisker orientation occurred, and the whiskers were misaligned. Moreover, some whiskers could not bear higher load and inevitably produce fracture in a whisker length distribution. Here, based on the real whisker orientation distribution in ABOw/Al–12Si (see Figure 1), $\lambda_{1}=\;0.94$ and $\lambda_{1}\lambda_{2}=\;0.65$ are reasonable values for evaluating the effect of aluminum borate whiskers on creep rates. Incorporation of the above values into Equation (5) helped obtain the value of $\frac{{\dot{\varepsilon }}_{c}}{{\dot{\varepsilon }}_{m}}=2.1\;\times 10^{-3}$, which is nearly closer to the experimental value of $\frac{{\dot{\varepsilon }}_{c}}{{\dot{\varepsilon }}_{m}}=2.18\;\times 10^{-3}$ under a stress of 70 MPa at 350 °C for the present ABOw/Al–12Si composite and the Al–12Si alloy. The theoretical results were verified by the experimental data, which showed that the addition of aluminum borate whiskers could decrease creep rate by nearly three orders of magnitude.

### 3.6. Creep Fracture Behavior

Figure 10 demonstrates the tensile fractographs of the Al–12Si alloy at 350 °C and 70 MPa. From Figure 10a, it can be observed that the overall morphology of the fracture, and its surface was very rough, showing obvious ductile fracture characteristics. An obvious tearing edge on the local area was further expanded, displayed in Figure 10b. The feature left after the silicon phase or intermetallic compound phase was pulled out from the aluminum matrix due to the brittle crack. In addition, there were cracks on the black bulk primary silicon. Based on the difference between the brittleness of silicon and the toughness of the aluminum matrix, it was more obvious at high temperature. Deformation inconsistencies existed between the silicon phase and the aluminum alloy matrix, when the material deformed under the action of external force, which caused stress concentration near the silicon phase. Thus, the cracks were often separated from the coarse silicon. The fracture morphology and mechanism are similar to the typical creep fracture of the aluminum alloy, as described in [10].

The fracture of the whisker-reinforced composite exhibited a macroscale brittle feature, unlike the ductile fracture of aluminum alloys. Figure 11 shows the typical creep fractographs at the test temperature of 300, 350 and 400 °C, respectively.

As shown in Figure 11a, at low temperature and high stress, the fracture surface of the composite showed obvious brittle fracture characteristics, and there were many small gray facets on the fracture surface. The EDS analysis presented that these facets resulted from the debonding between intermetallic compounds and the aluminum matrix [35,36]. A small amount of broken reinforcing whiskers can be observed, indicating that it is easier to engender fracture whiskers rather than debonding whiskers on the fracture surfaces. As the temperature rose to 350 °C, the creep fracture surface showed a great quantity-dimpled appearance after plastic deformation, indicating that the fracture surface was an obvious ductile fracture on the whole. Meanwhile, as described in Figure 11b, it can be found that the creep fracture surfaces consisted of the interfacial debonding and the fracture of whiskers. Figure 11c,d shows the fracture surfaces at 400 °C at different magnifications. When the creep temperature was 400 °C, the interfacial strength was so poor that the interfacial debonding was more likely to happen, as a great of traces left after the interfacial debonding are shown in the fractographs (Figure 11d). Moreover, the fracture surface tended to exhibit a macroscopically brittle fracture on the whole and a microscopically ductile fracture in the local region. As shown in Figure 11c, the number of dimples increased significantly, the size was larger and deeper, and the remaining holes were relatively rougher due to the pull-out of the whiskers. The ductile dimples and “tear ridges” might be attributed to the plastic deformation and fracture of the aluminum matrix softened at high temperature. The occurrence of interfacial debonding meant that the interfacial strength in the ABOw/Al–12Si composite was lower at 400 °C. Due to debonding, voids were formed and stress concentration was produced, resulting in the decrease of creep resistance, which is in agreement with the previous study [37,38].

Based on the above analysis, we figured out a model of composite fracture. Figure 12 presents the fracture model schematically. When the composite material was deformed at 300 °C, the strength of the matrix was very high and the matrix had a great constraint on the rotation of the whisker. If the rotation of the whisker was not coordinated with the deformation of the matrix, a quite serious deformation zone was generated around the whisker and the stress was easily concentrated around the whisker. When the stress accumulated to a certain extent, it led to the whisker fracture, as shown in Figure 12b. While at 350 °C, the rotation of the whisker was more coordinated with the deformation of the matrix owing to the resulting reduction of the matrix strength. Thus, the probability of the whisker fracture was reduced, when the composite was deformed at higher temperature. 

As the testing temperature increased to 400 °C, the matrix was more softened and basically did not restrict the rotation of the whisker, which resulted in the interfacial bonding strength weakening. Therefore, the reinforcing whiskers were not easy to fracture, and the matrix had to generate plastic flow deformation to accommodate the stress, which led to voids nucleation concentrated in the matrix. The interfacial properties of the composites were decreased at high temperature, which allowed the longitudinal whisker to be pulled out without rotation, and the transverse whisker was directly torn from the matrix, as shown by many traces of the pull-out and debonding of whiskers in the fractographs. Consequently, the cracks were more likely to initiate from the near-interface region and propagate along the interface to the matrix, leading to a fracture of the composite.

## 4. Conclusions

In the present research, the creep properties of ABOw/Al–12Si composites were investigated by means of a uniaxial tensile creep test in the temperature range from 250 to 400 °C and a stress range from 50 to 230 MPa. The conclusions are obtained as follows:(1)The analysis of the ABOw/Al–12Si composite creep data showed that this creep mechanism was dislocation climb (n = 4.03–6.02) with a creep activation energy of 148.75 kJ/mol. The dislocation climb in this composite was the dominant creep mechanism.(2)Static creep measurements results indicated that this composite has a very long stationary creep stage, compared to the unreinforced aluminum matrixes. Furthermore, the creep rate of the ABOw/Al–12Si composite was increased by about three orders of magnitude compared to that of the Al–12Si alloy under same testing condition. This indicated that the creep resistance of the composite has been greatly improved by adding whiskers.(3)The threshold stresses of the ABOw/Al–12Si composite were determined by the extrapolation technique, exhibiting a strong temperature dependence. For the static creep, a stress exponent of 4 was calculated by the means of introducing threshold stress into the analysis, and the threshold stresses decreased by improving the creep temperature from 300 to 400 °C.(4)A load transfer model was introduced to interpret the contribution of whiskers to the creep resistance of the composite. The experimental results values are well consistent with the results obtained from the creep rates based on the load transfer model.(5)The creep fracture surface of the ABOw/Al–12Si composite indicated that the main fracture mechanism is reinforcing whiskers rather than debonding whiskers at 300 °C whereas the dominant fracture mechanism at 400 °C is the interfacial debonding between the whiskers and the matrix. Meanwhile, the fracture surfaces consisted of interfacial debonding and fracture whiskers at 350 °C.

## Figures and Tables

**Figure 1 materials-14-01217-f001:**
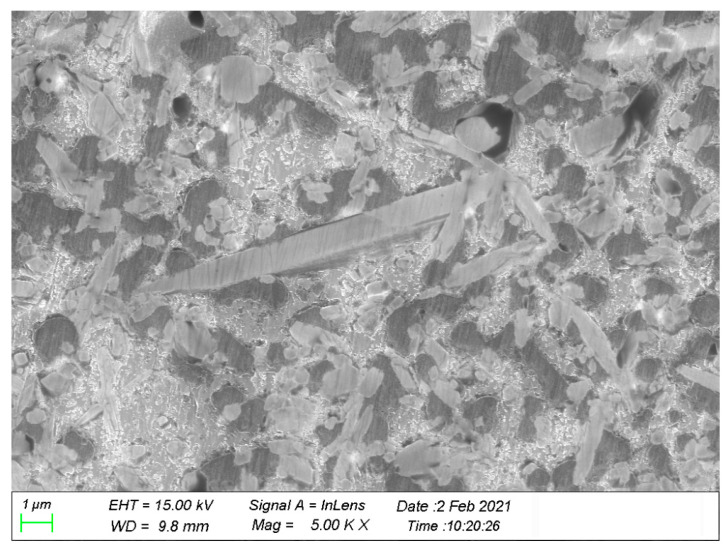
SEM fractograph of the ABOw/Al–12Si composite.

**Figure 2 materials-14-01217-f002:**
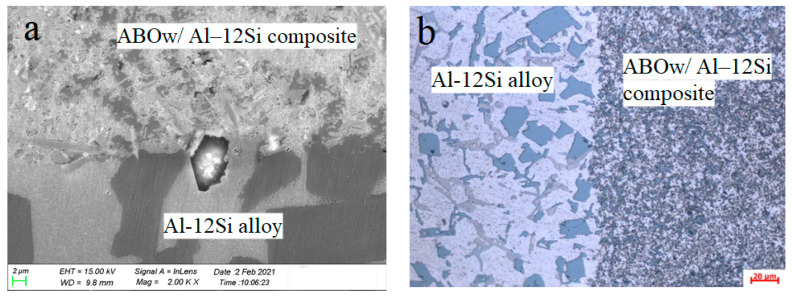
The (**a**) SEM fractograph with a 2000 magnification; and (**b**) optical micrograph with a 500 magnification of the composite bonding zone between the ABOw/Al–12Si composite and the Al–12Si alloy.

**Figure 3 materials-14-01217-f003:**
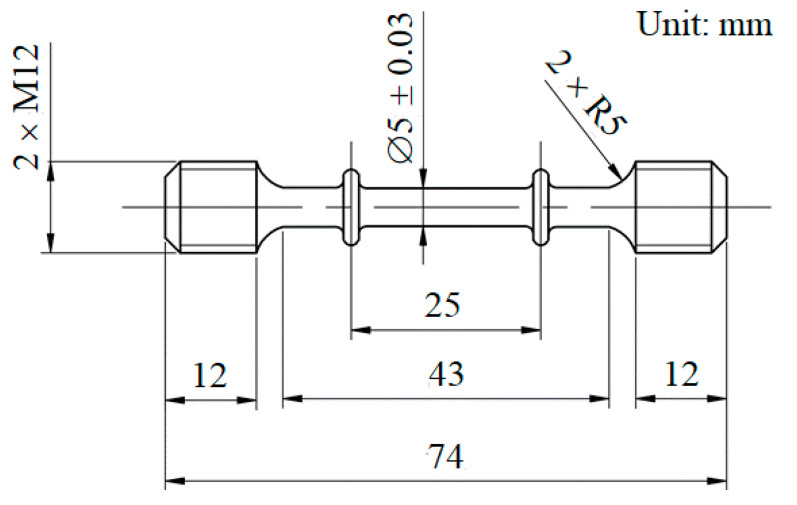
Dimensions of tensile creep specimens.

**Figure 4 materials-14-01217-f004:**
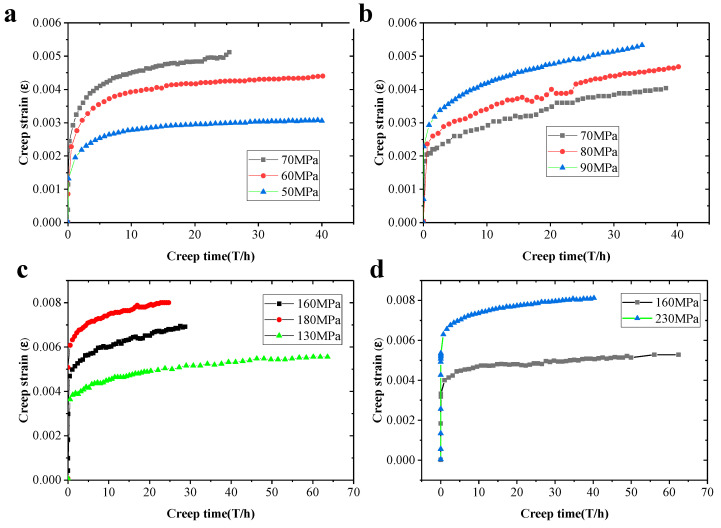
Tensile creep curves for the ABOw/Al–12Si composites tested at 400 °C (**a**), 350 °C (**b**), 300 °C (**c**), and 250 °C (**d**).

**Figure 5 materials-14-01217-f005:**
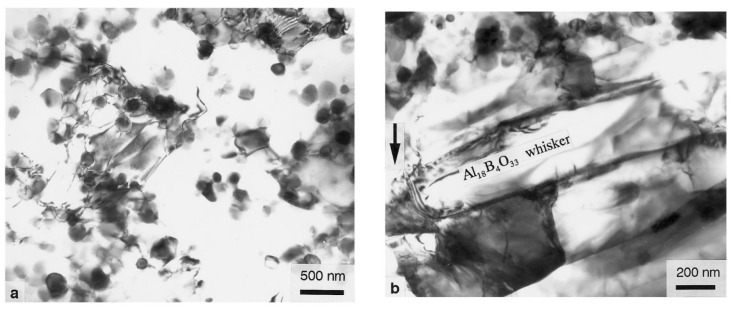
(**a**) TEM photographs showing dispersoids with dislocations. (**b**) AlBO whiskers after a creep test in [18,19].

**Figure 6 materials-14-01217-f006:**
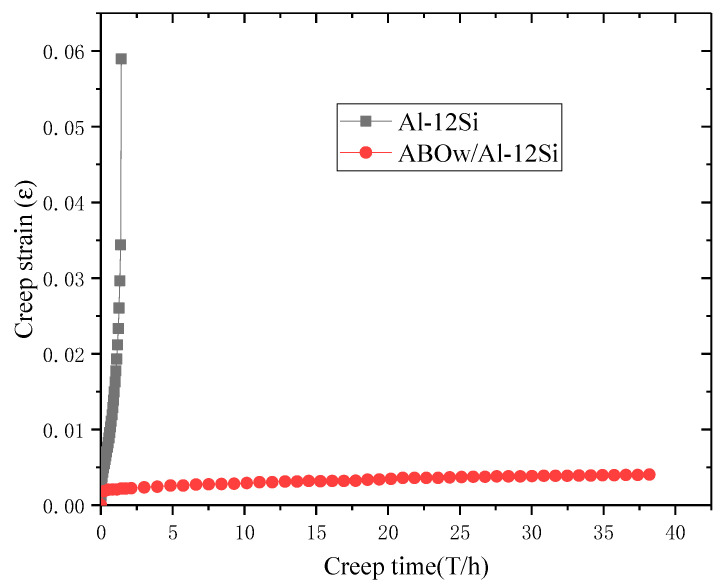
Comparison of the creep curves for the ABOw/Al–12Si composites and the Al–12Si alloy at 350 °C under an applied stress of 70 MPa.

**Figure 7 materials-14-01217-f007:**
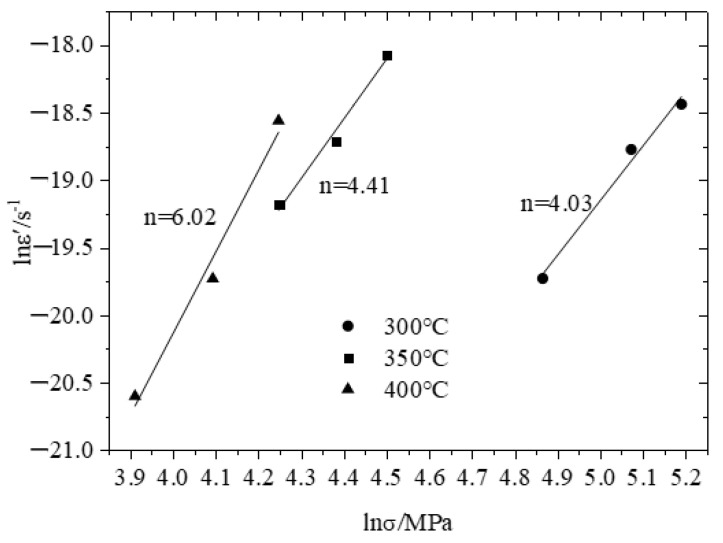
Plot of ln$\dot{\varepsilon }$ vs. lnσ for calculating the stress exponent for the ABOw/Al–12Si composite.

**Figure 8 materials-14-01217-f008:**
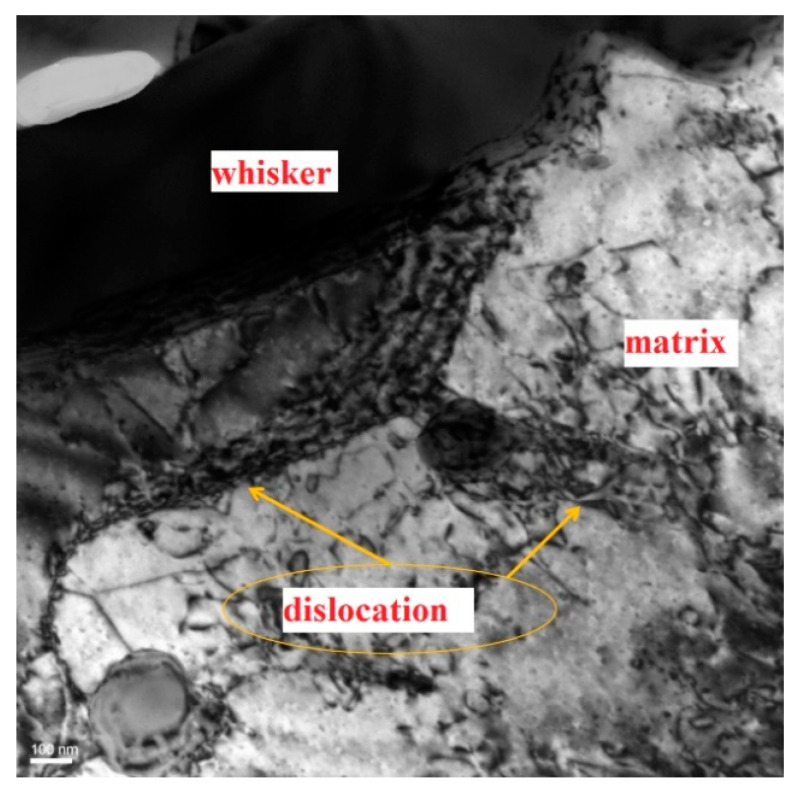
TEM image of microstructures after tensile creep deformation at 350 °C under 70 MPa.

**Figure 9 materials-14-01217-f009:**
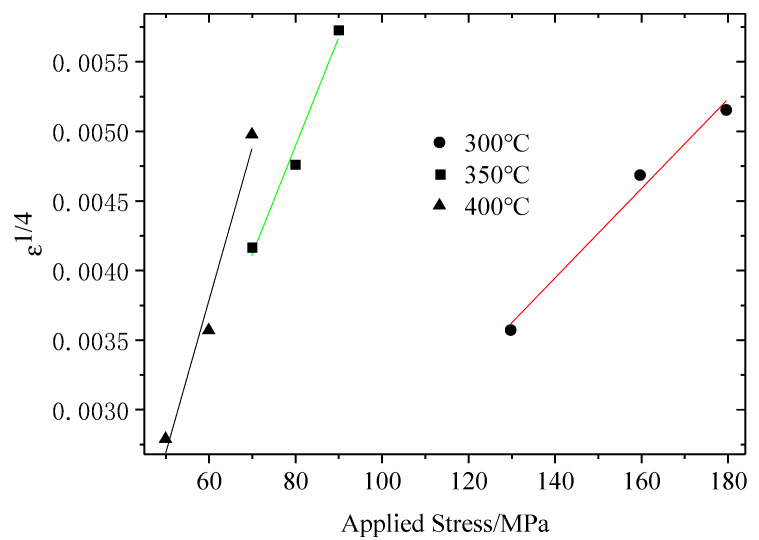
Calculation results of the threshold stress for the creep of the ABOw/Al–12Si composite.

**Figure 10 materials-14-01217-f010:**
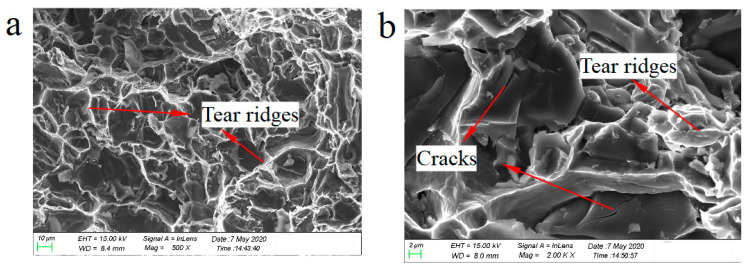
Tensile creep fracture surfaces of the Al–12Si alloy at different magnifications fractured at 350 °C and 70 MPa: (**a**) lower magnification; (**b**) higher magnification.

**Figure 11 materials-14-01217-f011:**
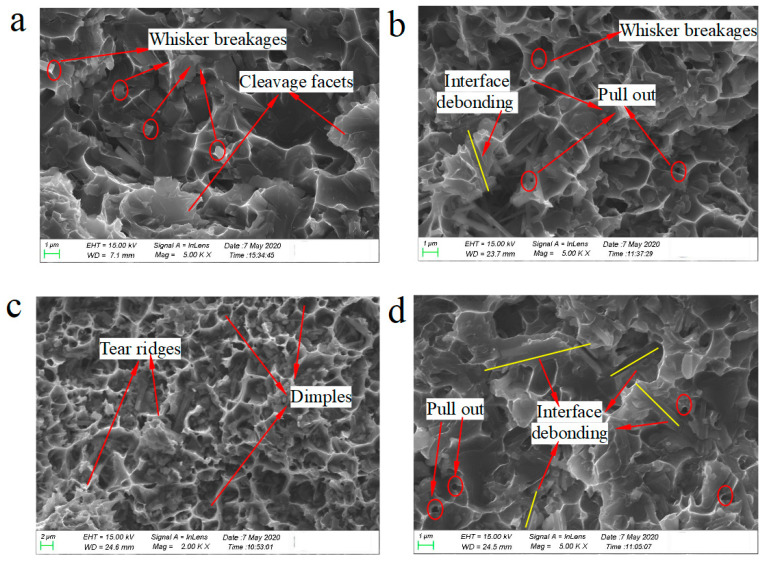
Scanning electron microscopy images of the ABOw/Al–12Si composite after creep at (**a**) 300 °C and 160 MPa, (**b**) 350 °C and 90 MPa, (**c**) 400 °C and 70 MPa (lower magnification), (**d**) 400 °C and 70 MPa (higher magnification).

**Figure 12 materials-14-01217-f012:**
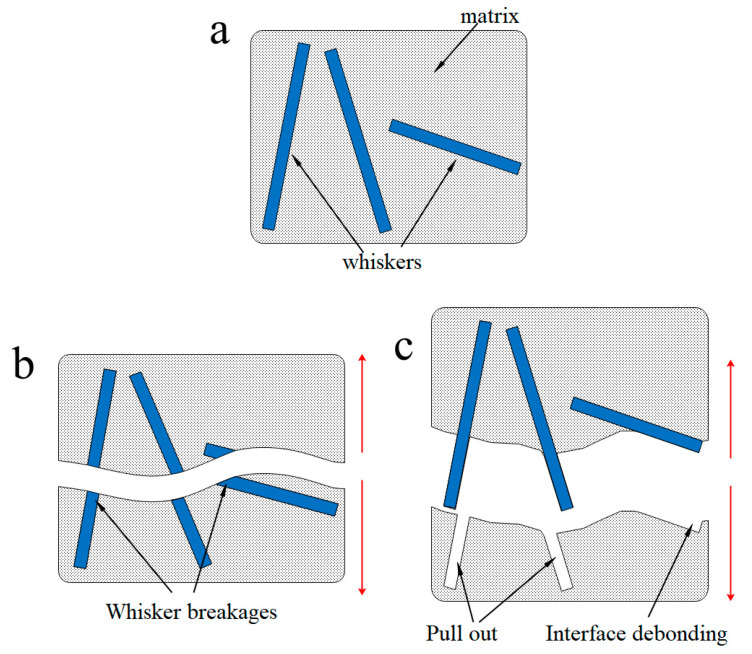
Schematics showing the creep fracture mode for the whisker-reinforced composite: (**a**) microstructure of the ABOw/Al–12Si composite before creep; (**b**) creep fracture mode after the tensile creep test at 300 °C; and (**c**) creep fracture mode after the tensile creep test at 400 °C.

**Table 1 materials-14-01217-t001:** Chemical composition of the Al–12Si alloy (wt.%).

Si	Cu	Ni	Mg	Fe	Mn	Ti	Zn	Al
12.47	4.21	2.97	0.80	0.38	0.25	0.112	0.007	Bal

**Table 2 materials-14-01217-t002:** Tensile properties of the ABOw/Al–12Si composite and the Al–12Si alloy at different temperatures.

Materials	Temperature (°C)	E (GPa)	UTS (MPa)	Elongation (%)
Al–12Si	25	80.3	312	3.3
	200	76.8	232	4.0
	250	67.6	166	4.8
	300	55.8	130	5.3
	350	54.6	94	6.0
ABOw/Al–12Si	25	113	394	0.61
	200	98	356	0.86
	250	87	327	1.01
	300	81	253	1.14
	350	75	217	1.25

E, Young’s modulus. UTS, ultimate tensile strength.

**Table 3 materials-14-01217-t003:** Creep properties of the alloys at different stresses and temperatures.

Temperature (°C)	Stress (MPa)	Steady-State Creep Rate (×10^−8^ s^−1^)
250	160	0.27639
	230	0.40833
300	130	0.27056
	160	0.94889
	180	0.98222
350	70	0.62917
	80	0.74417
	90	1.10056
400	50	0.11306
	60	0.26972
	70	0.86861

**Table 4 materials-14-01217-t004:** Values of stress exponent *n* for different composites.

Materials	Temperature (°C)	Stress Exponent
ABOw/Al–12Si	300	4.03
	350	4.41
	400	6.02
Al–12Si	350	4.18

**Table 5 materials-14-01217-t005:** Values of the threshold stress estimated by the extrapolation technique for the ABOw/Al–12Si composite.

Temperature (°C)	Threshold Stress (MPa)
300	37.41
350	25.85
400	17.36

## Data Availability

The data presented in this study are available on request from the corresponding author.

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
