# Peer review of "Elevated Temperature Tensile Creep Behavior of Aluminum Borate Whisker-Reinforced Aluminum Alloy Composites (ABOw/Al–12Si)"

_materials, 2021, doi:10.3390/ma14051217_

Round 1

Reviewer 1 Report

Reviewer Comments to Manuscript

 “Elevated temperature tensile creep behavior of aluminum borate whisker reinforced aluminum alloy composites (ABOw/Al–12Si)”

The paper topic is hot. Authors made interesting and important work. However, some paper aspects, statements and explanations need to be added, corrected, and edited.

The comments are below:

  1. Line 68: “The aim of the current study is to investigate the creep deformation properties of AlBOw reinforced Al matrix composites and compared with the unreinforced Al–12Si alloy” English needs to be corrected
  2. Line 74: Unfortunately, authors only give a grade of whiskers in the paper title. It is mentioned in experimental procedure that “… ABO whiskers have a length range of 10–30 um and a diameter range of 0.5–1.5 um”. However, no information about them (whiskers  structure, properties) is presented.  According to [14, 18], aluminum borate (Al18B4O33, denoted by ABO) whiskers are used for reinforcement. However, authors do not analyse this information and describe in detail the reinforcement phase being used in this work.
  3. Line 76:Authors analyse the Fig. 1 which is made at low magnification: “… It can be seen that the distribution of aluminum borate whiskers in  the composite is homogeneous and random, and without matrix cast defect”. However, it is impossible to see the mentioned features at this resolution. Authors need to present new images with high resolution and magnification.  
  4. Lines 79-82: Authors say: “…As shown in Fig. 2, obvious bulk primary silicon, aluminum and intermetallic compounds can be seen in the aluminum base alloy on the left side, most of which are hexagonal, petal and dendritic in shape. The average grain size measured is 32um. However, the average grain size of the  composite on the right side is 2.6um, which indicates that the addition of whiskers makes α-Al and  silicon particles further refined.” However it is impossible to define the grain size in the range of 1-2um at this image. Authors need to present new images with high resolution and magnification. 
  5. Line 102: 1 Optical micrographs of ABOw/ Al–12Si composite - small magnification. Authors need to increase the scale bar
  6. Line 116: Not proper English: “…With the increase of temperature from room temperature to…”
  7. Line 119: Not proper English: “…Moreover, the Young's modulus is the characterization of the material's ability to resist deformation”  -to correct
  8. Line121-123: The sentence “…Furthermore, the Elongation of Al–12Si alloy is the largest, while the Elongation of ABOw/Al–12Si composite is the smallest, revealing that the plastic properties of composites are poor compared with that of aluminum alloy from 25℃ to 350℃” is not clear . Authors need to correct English.
  9. Lines 125-126: Authors state: “…It is well known that reinforced whiskers, as barriers to hinder the dislocation movement, limit the deformation of matrix under applied load, resulting in the decrease of plasticity.” However, this statement is general and does not relate specifically to metal matrix composites (MMCs). Some citations need to be shown to demonstrate mentioned effects in MMCs. Moreover, other mechanisms such as the Harper-Dorn, dislocation, and power law breakdown [16], recrystallization, etc. need to be mentioned.
  10. Lines 145-156: Description at the lines 145-156 is repetition of explanations made in the works [17,18] without any micrographs and TEM images. So, authors need to add some TEM and other results of structure examination
  11. Lines 157-162: Not proper English (word “obvious” is used three times): “However, for the composite materials, there is no obvious accelerated creep stage in the creep fracture samples, which indicates that the creep fracture is sudden and the composite materials have obvious brittle fracture characteristics. Moreover, there is an obvious transition behavior between  the initial creep stage and the stationary creep stage, which is consistent with the description in [20]. Furthermore, it is obvious that the stationary creep stage arise at a shorter period of time and creep resistance decreases gradually, with the increase of creep stress under a constant temperature.” Please, correct.
  12. Lines187-188: The authors’ statement “…whisker affects the growth and refining of the aluminum alloy grains from Fig.2” cannot be based on micrograph of Fig.2 because of very low magnification and resolution of this optical micrograph. Authors need to present clear experimental evidence of grain growth. Secondly, the term “refining of the  aluminum alloy grains” is not clear and needs to be explained.
  13. Lines 188-196: Unfortunately, there is not any evidence of the dislocation structure modification processes described within the lines188-196. Authors need to present clear microstructure examination data proving this description.
  14. Line 208: “… s is the applied stress” – mistake : σ - is the applied stress
  15. Line 230: Equation (3) has a mistake - ln(ε1/ ε2). To correct: ln(ε′1/ ε′2).
  16. Lines 234-235: The statement “… Dislocation creep or power law creep mainly occurs by the movement of dislocations, which is helped by the dislocation climb aided by the diffusion of vacancies [24]” is not clear. English is not proper. What type of dislocation movement do authors keep in mind?
  17. Line 244: “…To reveal the operative deformation mechanism, the dislocation substructure developed in specimens after the minimum creep rate was reached was examined by TEM.” -  Not proper English
  18. Lines 250-252: The authors’ microstructure description “… Fig. 7 (a and b) is typical subgrain caused by dislocation in the sample under the applied stress 70 MPa at 350 ℃. Fig. 7 (a) shows the structural morphology of dislocations near the whisker, which shows that dislocations are accumulated near the whisker.” looks discussible. First of all, it is not completely true that  “ typical subgrain is caused by dislocation”.  It is well known the  cells (subgrains) are formed  by dislocation walls due to dislocation movement. Usually,  the cells are clearly seen on the TEM micrographs. Unfortunately the real cell (subgrain ) boundaries are not seen on the Fig.7a. Separate dislocations are seen, and they do not form dislocation configurations. It means that authors cannot see the subgrain structure and accumulation of dislocations near the whisker boundary. Moreover, authors did not examine the whisker-matrix interface. For this reason, it is not clear how to talk about dislocation accumulation at the whisker-matrix boundary. Annihilation of dislocations might be the possible mechanism in some cases
  19. Line 271: “… Therefore, (4) can be rewritten as” contains mistake. Are authors talking about Eq. (1)?  If so, authors missed in Eq. (4)  the part exp(-Q/RT) as shown in [12].
  20. Lines 293-294: Authors stressed attention without any explanations that “…The influence of aluminum borate whiskers in aluminum alloy on creep behavior are much not the same as particle-reinforced aluminum alloy”. So, authors need to explain this statement based on the differences of whisker-matrix interface structure
  21. Lines 392- 394. Authors discuss about a decrease of interfacial properties of the composites at high temperature: “… The decrease of interfacial properties of the composites at high temperature, which allowed the longitudinal whisker is pulled out without rotation, and the transverse whisker is directly torn from the matrix, as shown many traces of interfacial debonding and pull-out of whiskers in the fractography”. However, there is no any real data about evaluation of the interfacial properties of the composites at high temperature.
  22. Line 396: Conclusions need to be slightly corrected based on previous comments.

Author Response

Thank you for taking your time on reviewing our revised manuscript (ID: materials-1095911) entitled “Elevated temperature tensile creep behavior of aluminum borate whisker-reinforced aluminum alloy composites(ABOw/Al–12Si)”. The valuable suggestions are very helpful for revising and improving our paper. We have studied your comments carefully and have made some essential changes in our manuscript. The revised parts are marked in red in the revised manuscript.

Reviewer 2 Report

From a scientific point of view, the work presented in this paper is commendable. However, most of the research just confirm what other authors already did. It is not clear what is the scientific contribution of the present work (what is really new and what are the differences which can justify why the work was performed). At this point it is not clear if the work brings something newer and better (research work) or it is just a commendable, but routine experimental work, performed just to (double-) check and confirm what others did. I strongly recommend that, at least in the conclusion section, to stand out which results can be considered as contributions of the present work and which results just confirm what others did. As a general remark, the editing of the paper is very negligent, the imposed template of the journal, especially in terms of spaces (spaces between paragraphs, spaces between text and figures, spaces between text and tables) is often not adhered. There are tables and figures that are fragmented into two pages, which makes them very difficult to follow and read. Below are just some examples: Lines 31-32: “The strength and reliability of diesel engine combustion chamber parts are required to be higher” – higher than what? (consider replacing “higher” with “very high”) Lines 270-271: “Therefore, Eq. (4) can be rewritten as” … and equation 4 follows. Perhaps it was intended to write “Therefore, Eq. (1) can be rewritten as” References: have to comply to the journal template regarding the references style

Reviewer 3 Report

The paper is written in a logical way and in good English. However the some minor suggestions must be considered before publication:

  1. In chapter 2 should be given the international standard and designation of the Al-12Si alloy, e.g. EN 1676 or 1706, EN AB-47100. If the alloy has a non-standard chemical composition, the methodology for its implementation should be written.
  2. In Chapter 3 should be given estimate the errors of E, UTS,  Elongation in Table 2 and add error bars on Fig.4.

    Best Regards

Reviewer 4 Report

-the authors should clarify the novelty of this work in the introduction. 

-insert some recent application Al matrix composites as Progress in Materials Science 112 (2020) 100663.

-Could you please determine the difference between the two-phases in Fig.2? 

Round 2

Reviewer 1 Report

No comments

Reviewer 2 Report

The authors have addressed all reviewer's concerns. The paper is now worthwhile for publication.